# Detection of Low-Concentration Biological Samples Based on a QBIC Terahertz Metamaterial Sensor

**DOI:** 10.3390/s24113649

**Published:** 2024-06-04

**Authors:** Bing Dong, Bo Wei, Dongshan Wei, Zhilin Ke, Dongxiong Ling

**Affiliations:** 1School of Electrical Engineering and Intelligentization, Dongguan University of Technology, Dongguan 523808, China; dongbing2021@email.szu.edu.cn (B.D.); huilm980110@dgut.edu.cn (Z.K.); 2College of Electronics and Information Engineering, Shenzhen University, Shenzhen 518060, China; 3School of Physics and Optoelectronic Engineering, Yangtze University, Jingzhou 434023, China; weibo01.stu@yangtzeu.edu.cn; 4Shenzhen Institute of Advanced Technology, Chinese Academy of Sciences, Shenzhen 518055, China

**Keywords:** terahertz, metamaterial sensor, Quasi-bound state in the continuum, lithium citrate, bovine serum albumin, limit of detection

## Abstract

Quasi-bound state in the continuum (QBIC) can effectively enhance the interaction of terahertz (THz) wave with matter due to the tunable high-Q property, which has a strong potential application in the detection of low-concentration biological samples in the THz band. In this paper, a novel THz metamaterial sensor with a double-chain-separated resonant cavity structure based on QBIC is designed and fabricated. The process of excitation of the QBIC mode is verified and the structural parameters are optimized after considering the ohmic loss by simulations. The simulated refractive index sensitivity of the sensor is up to 544 GHz/RIU, much higher than those of recently reported THz metamaterial sensors. The sensitivity of the proposed metamaterial sensor is confirmed in an experiment by detecting low-concentration lithium citrate (LC) and bovine serum albumin (BSA) solutions. The limits of detection (LoDs) are obtained to be 0.0025 mg/mL (12 μM) for LC and 0.03125 mg/mL (0.47 μM) for BSA, respectively, both of which excel over most of the reported results in previous studies. These results indicate that the proposed THz metamaterial sensor has excellent sensing performances and can well be applied to the detection of low-concentration biological samples.

## 1. Introduction

Terahertz (THz) wave has great potential for substance detection by virtue of its broadband, low energy, and unique fingerprint spectrum [1,2,3,4]. However, with improvements in the required detection accuracy, the amount of reagents and the concentration of the test samples are often at the micro or even trace level, resulting in the traditional THz detection technology having problems such as weak response to substances and difficulty in capturing the spectrum. To address these problems, many structures and devices including micro- and nanofluidics [5,6,7], attenuated total reflection (ATR) [8], and micro- and nanostructures [9] have been proposed and applied to enhance THz spectral signals to some extent. However, the enhancement to THz spectroscopy detections of very low-concentration biological samples is still limited.

As an artificial composite material, metamaterials have unique properties which are not found in natural materials [10,11,12]. Metamaterials can interact with incident THz wave and stimulate the directional movement of surface electrons, so that the metamaterials can realize the enhancement of THz wave within the local area, and thus capture more information about the light–matter interaction [13]. THz metamaterial enhancement performances can be characterized by the Q factor and the sensitivity *S*. High Q factor and high sensitivity have been the unremitting pursuit in the field of THz spectroscopy detection. Conventional metamaterial design methods such as surface plasmon resonance (SPR) [14,15,16], Fano resonance [17], electromagnetically induced transparency (EIT) [18,19] and Mie resonance [20,21] have difficulty achieving both high Q factor and high sensitivity.

Bound state in the continuum (BIC) is a wave that remains localized even though it coexists with a continuous spectrum of the radiation wave that can carry energy away [22], i.e., it is an eigenmode without electromagnetic energy leakage [23,24], and hence its theoretical Q factor can reach infinity. While Quasi-bound state in the continuum (QBIC) makes a small portion of the electromagnetic energy leakage to the far field by introducing a structural asymmetry into the BIC mode [25], by controlling the asymmetry, it is possible to control the degree of electromagnetic energy leakage and thus actively regulate the Q factor [26,27,28], which provides an important idea of designing metamaterials sensors with adjustable high Q factors.

Since its potential feasibility of achieving both high Q factor and high-sensitivity detection of low-concentration samples, the THz metamaterial sensor based on QBIC has been a research hotspot recently. Some of representative works in the last two years are briefly introduced as follows. In 2023, Chang and Du et al. [29] presented a chip-based portable ultra-sensitive THz metasensor comprised of the split-ring resonator metasurface, which supports magnetic dipole QBIC, combined with functionalized gold nanoparticles (AuNPs) and used this metasensor to detect ultralow concentration of C-reactive protein and Serum Amyloid A protein down to 1 pM. Wang et al. [30] proposed a distinctive 2.5D out-of-plane architecture based plasmonic symmetry protected (SP)-BIC metasurfaces and demonstrated a detection limit of endotoxin as low as 0.01 EU mL^−1^. Liu et al. [31] manipulated the interference coupling between electric quadrupole and magnetic dipole in the metallic metasurface to excite ultrahigh quality QBIC with the Q factor of up to 503 and used this metasurface to detect trace homocysteine (Hcy) molecules to obtain a limit of detection of 12.5 pmol/μL, which is about 40 times better than that of the classical Dipole mode. In 2024, Peng et al. [32] designed and fabricated a structure-engineered THz QBIC metamaterial sensor consisting of a “double C” array, in which an asymmetry parameter α is introduced into the structure by changing the length of one subunit. They utilized the sensor to detect the arginine molecules and obtained a limit of detection of 0.36 ng/mL. Zhong et al. [33] prepared a THz QBIC metamaterial sensor based on an all-silicon dielectric material with periodically etched grooves and holes on the surface and obtained a QBIC mode with a high Q factor. Using the sensor to detect amino acid molecules, they obtained the lowest detection quantitation of 8.7 nmol. Huang et al. [34] reported a novel terahertz (THz) biosensor based on the integration of QBIC with graphene to discern concentrations of ethanol and N-methylpyrrolidone in a wide range from 100% to 0%, with the lowest detection concentration value of 0.21 pg/mL.

In this paper, a novel THz metamaterial sensor comprised of a dual-chain separated resonant cavity structure on a substrate based on QBIC is designed and fabricated and its sensing performances are analyzed and tested in simulation and experiment. The rest of this paper is organized as follows. In Section 2, the THz metamaterial sensor is designed, and the related sensing performances are discussed by simulations. In Section 3, the sensor is fabricated, and the THz spectroscopy measurement method is introduced. In Section 4, the THz metamaterial sensing experiments of two biological samples are performed and the results are discussed. In Section 5, conclusions of this work are given.

## 2. Design and Sensing Performance Simulations

### 2.1. Structural Design of the Metamaterial Sensor

Since the double-chain structure is convenient for easily constructing two symmetric or asymmetric cavities with one cavity on one chain, which are necessary for generating the BIC or QBIC mode, the designed metamaterial sensor is comprised of a dual-chain separated resonant cavity structure and is shown in Figure 1. Its unit structure with two separated resonant cavities is pointed out by the blue arrow in Figure 1 with structural sizes of *P*_1_ = 40 μm, *P*_2_ = 80 μm, *l*_1_ = 3 μm, *l*_2_ = 28 μm, *d* = 30 μm, and *d*_1_ = 2 μm and the thickness of 200 nm. The substrate of the metamaterial sensor is high-purity quartz with a thickness of *W* = 500 μm. Symmetry-protected excitation of QBIC is broken by varying the magnitude of *d*_2_. 

All of the simulations in this work are performed with the CST Studio Suite 2022. The materials used for the structure are selected from the CST material library (SiO_2_, Au, and PEC). The periodic conditions are implemented in the x- and y- directions and the z-direction is set to open by adding a vacuum space. As shown in Figure 2a, under the excitation of the vertically incident THz wave, the upper and lower resonance cavities generate a pair of reverse currents (blue arrows), which produce a circular magnetic field (violet arrows). The electric field distribution at the openings is demonstrated in the dashed box of Figure 2a. The magnetic field generated by the upper and lower loops is coupled in the resonance cavities and their gap region, thus exhibiting a binding force on the photons and resulting in confinement of the light field in a large area around the loops, and the resonance cavities as shown in Figure 2b, which may ensure the sensor can capture more sample molecules on the surface and realize the trace detection of samples.

### 2.2. QBIC Characteristics and Performance Optimization

In order to verify the formation mechanism of QBIC modes, the metamaterial surface structure was simulated using a PEC material to eliminate the effects of the ohmic loss [35]. Keeping the width *d*_1_ constant and increasing the width *d*_2_ from 0.5 to 4.0 μm, the degree of asymmetry α = (*d*_2_ − *d*_1_)/2 is defined and varies from −0.75 to 1. 

When the THz wave radiates on the surface of the metamaterial sensor in the TM mode, variation in the transmission resonant peak with the degree of asymmetry is shown in Figure 3a. When α = 0, the resonant peak cannot be observed in the spectrum. At this time, the sensor structure is in the symmetry-protected BIC mode and the energy is bound in the near-field and not leaking into the free space. Therefore, the Q factor is infinite, which causes the resonant peak in the spectrum with an infinitely narrow linewidth to become unobservable. When α ≠ 0, a resonance peak with a narrow line width is observed near the BIC mode, and the line width gradually increases with the increase in |α| due to the fact that the introduction of α breaks the symmetry of the original structure, leading to the transformation of the BIC mode with no energy leakage into the QBIC mode with partial energy leakage. The magnetic field energy at one of the openings is monitored in both BIC and QBIC modes as shown in Figure 3b. No magnetic field energy is observed in the BIC mode, while magnetic field energy leakage is observed around the structure in the QBIC mode, which verifies the transformation from BIC into QBIC.

Meanwhile, the relationship between the Q factor and the degree of asymmetry α is explored. The transmission spectrum of the metamaterial sensor at each α can be fitted with the Fano formula [36]:$$T_{Fano}=t_{0}\times {\left|a_{1}+ia_{2}+\frac{b}{\omega -\omega_{0}+i\gamma }\right|}^{2}$$
where $T_{Fano}$ is the transmission intensity, a_1_, a_2_ are constant real numbers, $\omega_{0}$ and γ are the resonant frequency and damping rate, respectively, and the Q factor of the Fano resonance is calculated under different α according to Q = $\omega_{0}$/2γ. Variation in the Q factor with |α| is obtained as shown in Figure 3c. It can be seen that the closer |α| is to 0, the larger the Q factor is. Furthermore, the Q factors are approximately the same at the same |α|. Previous studies had confirmed that the relation between the Q factor and α is inversely quadratic [37,38] with Q = Q_1_|α|^−2^, where Q_1_ is an independent constant and depends on the topology and material properties of the metamaterial itself. Here, the Q factor is fitted with the equation of Q = 98.96|α|^−2^ with R^2^ > 0.999. This is in accordance with the Q factor variation rule for the excitation of the symmetry-protected BIC mode, proving that the QBIC mode is excited by symmetry-breaking.

In the above simulations, the ideal conducting material of PEC was chosen to verify the relationship between Q and α, while the effect of the ohmic loss on the resonant strength is neglected [35]. To investigate the ohmic loss of the metal structure sensor, keeping *d*_1_ = 2 μm constant, increasing *d*_2_ from 3 to 9 μm, i.e., α increasing from 0.5 to 3.5, variations in the resonant peak intensity with α for the PEC sensor and for the gold (conductivity: 4.1 × 10^7^ S/m) sensor are shown in Figure 4a,b, respectively, for comparison. From these two figures, we can see there is an obvious difference in the resonant peak between the PEC and metal sensors. To quantitatively show the ohmic loss on the sensor, the Q factors at different α are plotted for comparison as shown in Figure 4c. It can be seen that the Q factor decreases as α increases and the variations of Q vs. α for the PEC and metal sensors are not consistent due to the ohmic loss, especially when α is small. To determine the optimal structural asymmetry, a figure of merit (FoM) of the simulated sensor is defined as FoM = Q × H [31,39], where *H* is the depth of the resonant peak as shown in the inset of Figure 4d. The relation between the calculated FoM and α is shown in Figure 4d. It can be seen that at α = 2, FoM reaches the maximal value, so α = 2 is adopted as the structural asymmetry with *d*_1_ = 2 μm and *d*_2_ = 6 μm in the following simulations. At this time, the Q factor has about a 30% decrease due to the ohmic loss seen from Figure 4c. 

### 2.3. Sensing Performance Simulations

To characterize the sensing performances of the metamaterial sensor, transmission spectra under different thicknesses and refractive indices of the detected sample are simulated and shown in Figure 5a,c. It can be seen that in comparison with the resonant frequency of the blank sensor, the resonant frequencies of the sensor covered by the sample with different thicknesses and refractive indices have different redshifts. The reason is due to the fact that the increased thickness and refractive index of the detected sample will increase the equivalent capacitance of the sensor, which will result in the decrease in the resonant frequency according to the LC circuit principle. To quantitatively illustrate the effects of the thickness and the refractive index of the detected sample on the transmission property of the sensor, the relations between the resonant frequency shift Δ*f* and the thickness *h* are shown in Figure 5b. From this figure, we can see that Δ*f* first nonlinearly increases with the increase in *h* and then saturates at 20 μm. This is due to the fact that as the thickness of the covered sample increases to exceed 20 μm, the upper part of the sample exceeding the edge of the bound optical field cannot interact with the confined electromagnetic wave, and therefore the thickness sensitivity disappears. From Figure 5d, we can see that the resonant frequency shift Δ*f* increases linearly with the increase in Δ*n* at different sample thicknesses, where Δ*n* is the refractive index change relative to the air of *n* = 1.0. By defining the sensitivity of the sensor as *S* = Δ*f*/Δ*n*, it can be seen that the maximal *S* is calculated to be 544 GHz/RIU at the saturated covering thickness of 20 μm. Even at the smallest simulation thickness of *h* = 1 μm, S_1_ = 220 GHz/RIU as shown in Figure 5d, indicating that the sensor also has good sensing performance for ultra-thin object detection.

To evaluate the sensing performance of the designed metamaterial sensor, the S parameters for different sensors reported in recent literature are listed in Table 1 for comparison. It can be seen that the sensitivity of the designed sensor is excellent.

## 3. Experimental Methods

### 3.1. THz Spectroscopy Equipment

All THz spectroscopy measurements in this work were performed using a THz time-domain spectroscopy (THz–TDS) spectrometer (Advantest Photonix TAS7500SU, Advantest Corporation, Tokyo, Japan). During the THz spectroscopy measurements, a maximal delay time of 132 ps was set and a frequency resolution of 7.6 GHz was obtained. Under the setting, this spectrometer can stably produce a spectrum every 200 ms. The THz–TDS spectrometer is equipped with a dry air purge accessory with which the humidity of the detection chamber can be controlled at around 5%. All THz spectroscopy measurements were performed in transmission mode at a temperature of 23.0 ± 1.0 °C. 

### 3.2. Fabrication of Metamaterial Sensor

According to the above simulation results, the metamaterial sensor with optimal parameters was fabricated using a combination of photolithography and vapor deposition processes. In order to bond the metal layer and the substrate more tightly, 10 nm thick chromium is added between the 200 nm gold metal layer and the 500 μm quartz substrate as an adhesive. One of the fabricated metamaterial sensors is shown in Figure 6a. The THz transmission spectrum of the fabricated sensor is displayed in Figure 6b. For comparison, the simulation transmission spectrum of the blank sensor is also shown. It can be seen that the resonant peak of the fabricated sensor has a blueshift of about 80 GHz and its depth is smaller compared to the simulated resonant peak, which may be due to the fact that the actual ohmic loss of the metal is higher than the simulated one and the quality of the experimental resonant peak degrades because of the impact of the processing accuracy.

### 3.3. Biological Sample Preparation

Two typical biological samples were prepared and tested in this work. One is lithium citrate (chemical formula: Li_3_C_6_H_5_O_7_, molecular weight: 209.92), which is widely used in the clinical treatment of psychosis and bi-directional affective disorder [46]. The other is bovine serum albumin (BSA), which is a white to light yellow lyophilized powder with a molecular weight of 66.44 kDa and a solubility of 100 mg/mL. Both of these two samples with purity > 99.0% were purchased from Macklin corporation (Shanghai, China) without further purification.

To prepare the lithium citrate (LC) solutions with different concentrations, 0.2 g LC sample was weighed and dissolved in 100 mL of deionized water to obtain the concentration of 2 mg/mL. To obtain lower concentration samples, the 2 mg/mL solution was sequentially diluted to 1.0, 0.1, 0.01, and 0.0025 mg/mL. Similarly, to obtain the BSA solution samples, 0.1 g of BSA sample was weighed and dissolved in 100 mL of deionized water to obtain the concentration of 1.0 mg/mL, then the 1.0 mg/mL BSA solution was diluted to obtain 0.5, 0.25, 0.125, 0.0625, and 0.03125 mg/mL solutions.

To perform the THz spectroscopy measurements, one of the 20 μL liquid samples with different concentrations was first dropped using a pipette on the surface of the metamaterial sensor and evenly spread using a homogenizer. Then, drying of the liquid sample on the surface of the metamaterial sensor was executed in an oven at a temperature of 40 °C for about 5 h. Each sample was measured three times to minimize the experimental error.

## 4. Experimental Results and Discussion

### 4.1. Detection of Lithium Citrate Based on the Metamaterial Sensor

Transmission spectra of LC solutions at different concentrations are shown in Figure 7a. It can be observed that the central frequency of the resonant peak has a redshift as the concentration of the LC sample increases. To quantitatively describe the resonant frequency redshift, variation in the Δ*f* with the concentration is shown in Figure 7b and the relation between them can be linearly well fitted with an equation of y = 118.88x + 31.40 with R^2^ = 0.97. Due to the spectral resolution of the spectrometer (7.6 GHz), when the LC concentration is 0.0025 mg/mL, Δ*f* reduces to the equivalent value with the spectral resolution as shown in Figure 7b. Therefore, the limit of detection (LoD) of LC by the metamaterial sensor is determined to be 0.0025 mg/mL (~12 μM). In comparison to the recent detection result of 0.1 mM [47] for potassium citrate with the traditional THz metamaterial, the QBIC THz metamaterial has one order of magnitude improvement on the LoD. In addition, the liquid amount dropped on the sensor is only 20 μL, which is also significantly reduced compared with previous THz metamaterial detection [41], indicating that the QBIC metamaterial sensor in this work has excellent sensing performance.

### 4.2. Detection of BSA Based on the Metamaterial Sensor

The THz transmission spectra of the BSA solutions with different concentrations were measured and the results are shown in Figure 8a. Similarly, the central frequency of the resonant peak has a redshift as the concentration of the BSA sample increases. The relation between the central resonant frequency shift and the BSA concentration can be linearly fitted with an equation of y = 104.85x + 12.59 with R^2^ = 0.98 as shown in Figure 8b. Similarly, considering the spectral resolution of the spectrometer, the LoD of the metamaterial sensor for BSA solutions is 0.03125 mg/mL (0.47 μM) as marked in Figure 8b. To compare this LoD with previously reported detection results of 1.52 μM by a polarization-insensitive THz sensor [48], 1 μM by a asymmetric split resonator THz sensor [47], 0.53 μM by an all-metal THz sensor [49] and 0.15 μM by a SRRs THz sensor [50], the QBIC sensor is advantageous in sensitivity for low-concentration protein detection. 

## 5. Conclusions

In summary, we designed and fabricated a THz metamaterial sensor based on the bound state in the continuum with a double chain-type separated resonance cavity structure. By controlling the size of the openings, an asymmetry is introduced into the structure. The distribution of electric and magnetic fields on the surface of the metamaterial sensor was simulated and the introduction of the asymmetry degree exciting the mode transformation from BIC to QBIC was verified. The thickness and refractive index sensitivity of the sensor were investigated and the saturated sample thickness and the highest refractive index sensitivity of the sensor were determined to be around 20 μm and 544 GHz/RIU, respectively, by simulations. In addition, the effect of ohmic loss on the metamaterial sensor was also discussed. Since the ohmic loss of the gold metal, the Q factor of the sensor decreases 30% compared with the PEC material. In the experiment, the detection capability of the metamaterial sensor for low-concentration biological samples was verified. For lithium citrate solution detection, the LoD was determined to be 0.0025 mg/mL (12 μM), which is one order of magnitude lower than the previous report. For BSA solution detection, the LoD was 0.03125 mg/mL (0.47 μM), greatly lower than the reported results using traditional THz sensors. Meanwhile, the amount of liquid required for a single detection is only 20 μL, also much lower than the requirements in the previous studies. Therefore, the proposed THz metamaterial sensor based on the QBIC mode has excellent sensing sensitivities and can be used for detection of low-concentration biological samples.

## Figures and Tables

**Figure 1 sensors-24-03649-f001:**
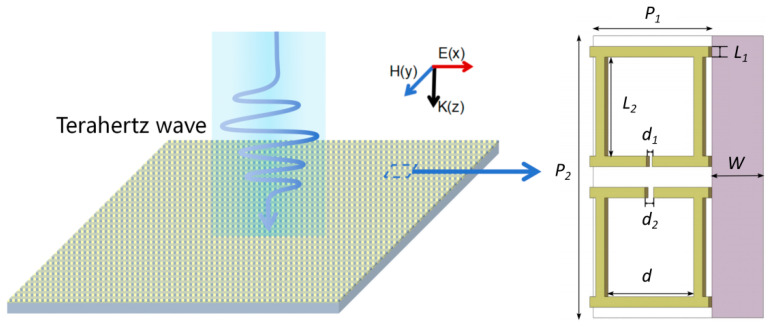
Structure of the sensor.

**Figure 2 sensors-24-03649-f002:**
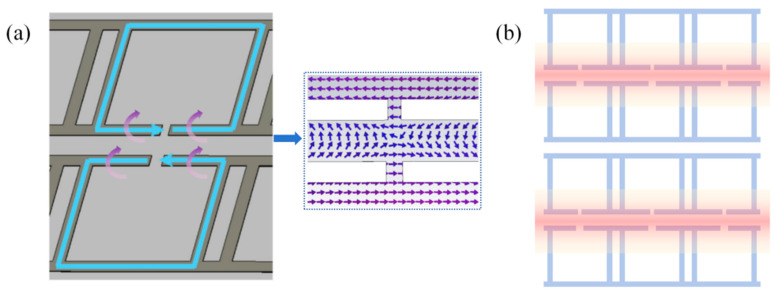
(**a**) Current and electromagnetic fields on the surface of the sensor; (**b**) Schematic of the constrained light field.

**Figure 3 sensors-24-03649-f003:**
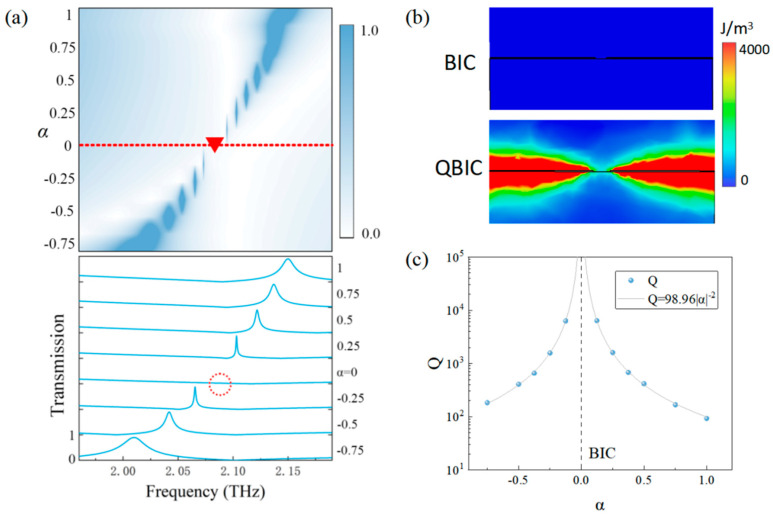
(**a**) The evolution of the resonance peak; (**b**) Magnetic field energy at one of the openings; (**c**) Q varies with α. The dotted line is for the eye guide of α=0 and the dotted circle and the triangle are marks of the BIC in (**a**).

**Figure 4 sensors-24-03649-f004:**
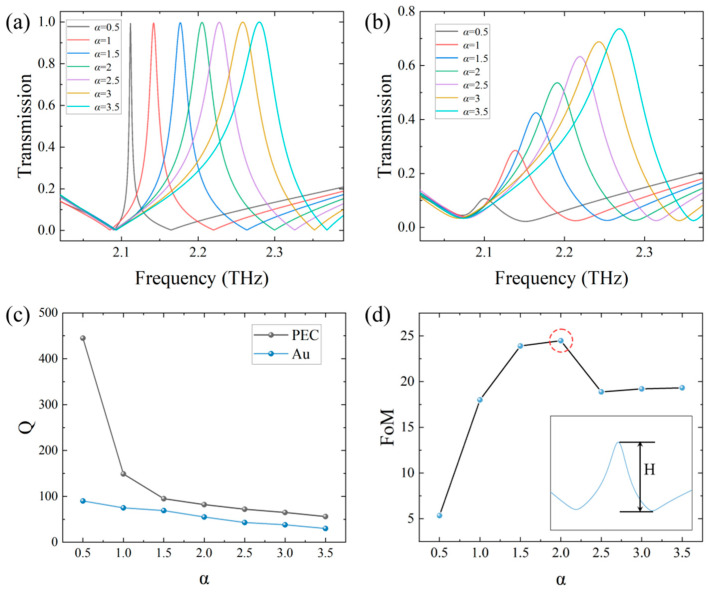
(**a**) Resonant peak variation with *α* using the PEC material; (**b**) Resonance peak variation with *α* using the gold metal material; (**c**) Variation in Q factor with *α* for different materials; (**d**) Variation in FoM with *α* and the definition of H in the inset.

**Figure 5 sensors-24-03649-f005:**
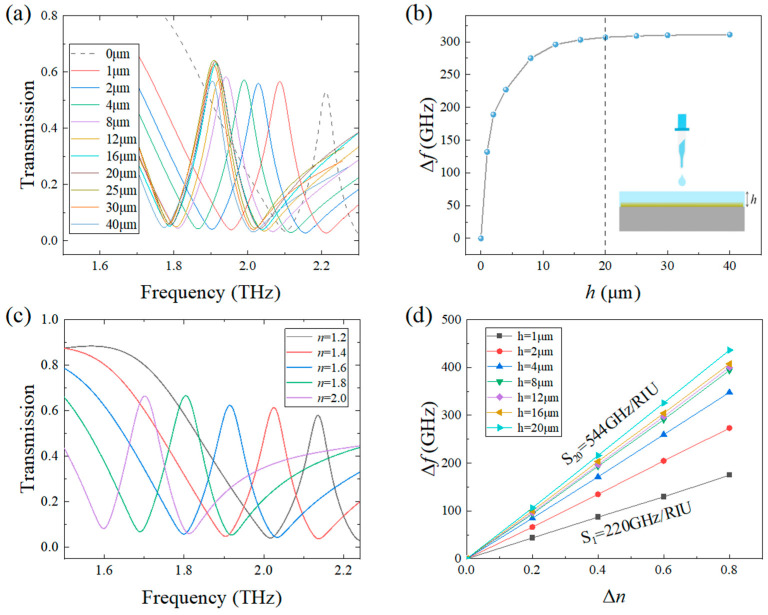
(**a**) Frequency shift varies with thickness of the detected sample; (**b**) relations between the resonant frequency shift and the thickness; (**c**) Frequency shift with refractive index at *h* = 20 μm; (**d**) Refractive index sensitivity at different thicknesses.

**Figure 6 sensors-24-03649-f006:**
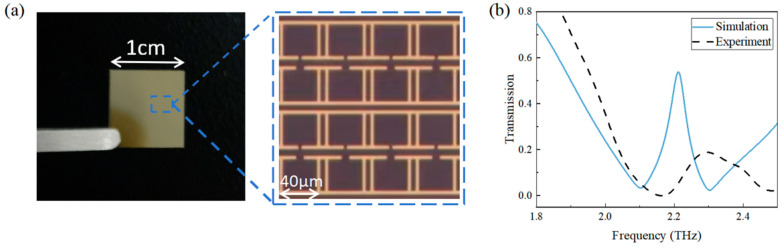
(**a**) Photograph of the metamaterial sensor; (**b**) Simulation and experimental spectra of the blank metamaterial sensor.

**Figure 7 sensors-24-03649-f007:**
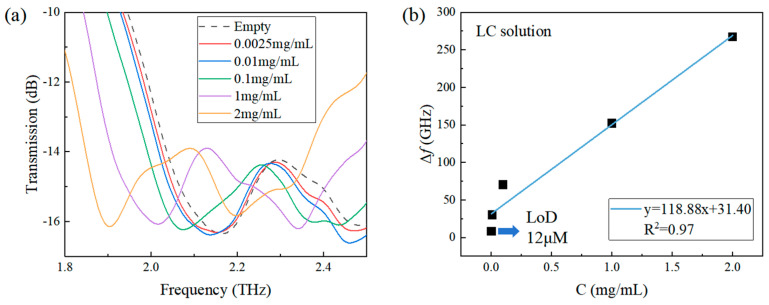
(**a**) THz transmission spectra of LC solutions at different concentrations; (**b**) Variation in the Δ*f* with the concentration. The error bar is not shown in (**b**) for clarity since it is smaller than or comparable to the size of the symbol of the data point at each concentration.

**Figure 8 sensors-24-03649-f008:**
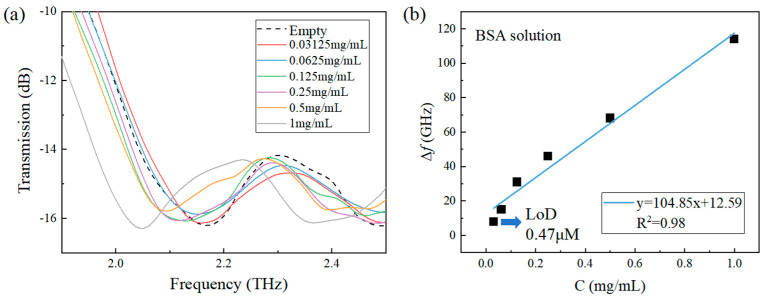
(**a**) THz transmission spectra of BSA solutions at different concentrations; (**b**) Variation in Δ*f* with the concentration. The error bar is not shown in (**b**) for clarity since it is smaller than or comparable to the size of the symbol of the data point at each concentration.

**Table 1 sensors-24-03649-t001:** Simulated refractive index sensitivity of the metamaterial sensor.

Structures	S (GHz/RIU)	Reference
Metal rings and double “I” cross structure	300	[40]
Asymmetric open ring	328	[41]
Gold nanoparticles and gold wires	123.45	[42]
QBIC-Fano Resonance	165	[43]
Double elliptic QBIC structure	293	[44]
Periodic array of two ring chain resonators	420	[31]
Square ring with T-shaped strips	37	[45]
Double-chain resonant cavity QBIC	544	This work

## Data Availability

Data underlying the results presented in this paper are not publicly available at this time but may be obtained from the authors upon reasonable request.

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
