# Peer review of "Detection of Low-Concentration Biological Samples Based on a QBIC Terahertz Metamaterial Sensor"

_sensors, 2024, doi:10.3390/s24113649_

Round 1
Reviewer 1 Report
Comments and Suggestions for Authors
The authors proposed a novel THz metamaterial sensor with a double-chain separated resonant cavity structure based on quasi-bound state in the continuum (QBIC) to high sensitively detect low-concentration biological solution samples. The sensitivity of the proposed sensor is 544 GHz/RIU in theory, which is much higher than those of recently reported THz metamaterial sensors and its sensitivity was validated by measuring two typical biological solution sample with a THz spectroscopy. This manuscript is well-organized and the results are clear and reasonable. The manuscript can be accepted after minor revision.
(1) The Fano formula is not clearly described. This formula should have been used to fit the transmission spectrum of the blank metamaterial sensor near the transmission peak to obtain the fitting parameters of omega_0 and gammar. In addition, the parameter T_Fano is not defined.
(2) In Figs. 7b and 8b, the error bars should be added.
(3) Some latest and related literatures are suggested to be added, e.g., Nanoscale, 2023, 15, 3398-3407; Nanomaterials 2024, 14(9), 799.
Comments on the Quality of English LanguageThis manuscript is well-organized and the results are clear and reasonable and it can be accepted after minor revision.
Reviewer 2 Report
Comments and Suggestions for Authors
In this study, the authors investigate a THz metamaterial sensor based on QBIC. While the research fits well within the journal's scope, and both simulation and experimental results show promise, there are several notable issues that require attention (minor revisions):
1. Please explain the reason and mechanism behind using the configuration of a metal double-chain for the proposed metamaterial sensor.
2. To enhance reader comprehension, elaborate in more detail on the settings used in the CST simulations.
3. In reference to Figure 4(c), mention the permittivity of gold used in the CST simulation. Additionally, explain why there is a significant difference in the resonant peak between the PEC and metal sensors when α=0.5.
4. Line 160 gives the definition of the figure of merit (FOM) as FOM = Q × H. As it is well known, the FOM is typically defined as FOM = Sensitivity/FWHM. Please elucidate this point in the text or cite a related reference.
5. Improve the references in the introduction section by incorporating recent related works on metamaterial sensors, specifically Plasmonics 19 (1), 481-493 (2024) and Plasmonics 18 (4), 1581-1591 (2023).
6. Meticulously review the manuscript to rectify any typographical errors (e.g., line 155, 4.1×10^7) and grammatical errors.
